# Uptake of Radionuclides by Bryophytes in the Chornobyl Exclusion Zone

**DOI:** 10.3390/toxics11030218

**Published:** 2023-02-25

**Authors:** Brigitte Schmidt, Felix Kegler, Georg Steinhauser, Ihor Chyzhevskyi, Sergiy Dubchak, Caroline Ivesic, Marianne Koller-Peroutka, Aicha Laarouchi, Wolfram Adlassnig

**Affiliations:** 1Core Facility Cell Imaging and Ultrastructure Research, Faculty of Life Sciences, University of Vienna, Djerassiplatz 1, 1030 Vienna, Austria; 2Institute for Physics, Martin Luther University Halle-Wittenberg, 06120 Halle, Germany; 3Institute of Radioecology and Radiation Protection, Leibniz University Hannover, Herrenhäuser Straße 2, Building 4113, 30419 Hannover, Germany; 4Chemistry & TRIGA Center Atominstitut, TU Wien, Getreidemarkt 9/163, 1060 Vienna, Austria; 5State Specialized Enterprise “Ecocentre” (SSE “Ecocentre”), 07270 Chornobyl, Ukraine; 6Functional and Evolutionary Ecology, Faculty of Life Sciences, University of Vienna, Djerassiplatz 1, 1030 Vienna, Austria

**Keywords:** americium, cesium, strontium, fallout, bryophytes, mosses, radioactivity

## Abstract

The “Chernobyl nuclear disaster” released huge amounts of radionuclides, which are still detectable in plants and sediments today. Bryophytes (mosses) are primitive land plants lacking roots and protective cuticles and therefore readily accumulate multiple contaminants, including metals and radionuclides. This study quantifies ^137^Cs and ^241^Am in moss samples from the cooling pond of the power plant, the surrounding woodland and the city of Prypiat. Activity concentrations of up to 297 Bq/g (^137^Cs) and 0.43 Bq/g (^241^Am) were found. ^137^Cs contents were significantly higher at the cooling pond, where ^241^Am was not detectable. Distance to the damaged reactor, amount of original fallout, presence of vascular tissue in the stem or taxonomy were of little importance. Mosses seem to absorb radionuclides rather indiscriminately, if available. More than 30 years after the disaster, ^137^Cs was washed out from the very top layer of the soil, where it is no more accessible for rootless mosses but possibly for higher plants. On the other hand, ^137^Cs still remains solved and accessible in the cooling pond. However, ^241^Am remained adsorbed to the topsoil, thus accessible to terrestrial mosses, but precipitated in the sapropel of the cooling pond.

## 1. Introduction

The explosion of the RBMK-1000 reactor in Unit 4 of the Chornobyl nuclear power station on April 26, 1986 is regarded as the worst nuclear accident in history, the “Chernobyl nuclear disaster”. A significant part of the inventory of the reactor was released, including 8.5 × 10^16^ Bq ^137^Cs with a half-life (T_1/2_) of 30.1 years, corresponding to approximately 33% of the inventory, 1.0 × 10^16^ Bq ^90^Sr with a T_1/2_ of 28.8 years, corresponding to approximately 4.3% of the inventory, and 4.2 × 10^12^ Bq ^241^Am with a T_1/2_ of 432.3 years. However, the amount of ^241^Am exceeds the comparatively small amount originally released, as its environmental concentrations have been increasing since the disaster, as ^241^Am is constantly produced by the decay of ^241^Pu. An amount of 6.0 × 10^15^ Bq ^241^Pu with a T_1/2_ of 14.3 years was set free, corresponding to approximately 3.3% of the inventory. In addition to these isotopes, numerous other radionuclides were released, most of which were rather short-lived [1]. Thus, the total activity decreased rapidly, but not in a linear manner, with <1% remaining in the global environment after ten years and <0.1% after 100 years [1]. A significant proportion of the radioactivity was released in small particles, which were predominantly deposited within 30 km of the reactor, mainly north and west of the reactor. Within this 30 km zone, deposition of ^90^Sr varied from 20 kBq/m^2^ to 20,000 kBq/m^2^, with even higher values at certain spots [1].

In the affected areas, plants were exposed to radioactivity by two distinct pathways:(1)Immediately after the incident, deposition of contaminated dust or precipitation led to the exposure of parts above the ground. Thereby, exposure rates were high enough to kill coniferous trees in the particularly affected “Red Forest” [2] and to cause pathologic leaf-fall and morphological abnormalities in deciduous trees [3].(2)After approximately one year, most radioisotopes mitigated to the soil. Subsequently, plants may have absorbed radioisotopes via their root systems, especially if the chemical properties of the radioisotopes were similar to essential nutrients, as in the case of ^137^Cs and potassium. This uptake was more pronounced on the sandy podzol soils in the close vicinity to the plants due to their weaker ion-binding capacity [1].

Bryophytes (mosses/Bryopsida and liverworts/Marchantiopsida and Jungermanniopsida) are known to be more resistant to radioactivity compared to “higher”, i.e., vascular plants [3,4], and tend to absorb much higher amounts of radionuclides [4,5]. Due to the wide geographical distribution of some species, mosses have even been used to map deposition of radioisotopes [6]. However, compared to fungi [7], conifers [8] and edible berries [2], wild-growing mosses in the Chornobyl exclusion zone have received comparatively little attention so far.

The high accumulation of radioisotopes by mosses is predominantly caused by two factors: mosses are usually small, with leaflets consisting only of a single cell layer. Thus, they have a very high ratio of surface to volume. Furthermore, the gametophyte, i.e., the green moss plant, lacks the impermeable cuticle present in vascular plants. The sporophyte, i.e., the spore capsule and its stalk, may possess a cuticle but contribute little to the total biomass of the plant. Mosses therefore absorb water and nutrients throughout their whole surface. On the other hand, mosses lack roots and are therefore not capable of extracting nutrients from deeper soil layers. Instead, mosses possess rhizoids, which predominantly fix the plant on its substrate and may also contribute to absorption, but only from the very surface of the substrate. The possibility of transport of absorbed elements within the moss plant, from rhizoids to leaflets, depends on the presence of vascular tissue. Some primitive Bryopsida (Polytrichaceae) exhibit vascular tissue similar to higher plants, but have a lower transport capacity. In more advanced Bryopsida, the vascular tissue is reduced and probably not functional anymore. In liverworts, no vascular tissue exists.

Thus, uptake of radioisotopes from the environment in mosses differs significantly from higher plants, both in quantitative and in qualitative terms. Furthermore, it can be expected that mosses will exhibit highly diverse patterns of uptake, depending both on their anatomy and the availability of radioisotopes at the very surface layer of the substrate (mainly soil, rock or tree bark, but also including exotic substrates such as animal corpses or hairballs) and possibly also in dust deposition. A study on ^137^Cs in Turkish Bryophytes found pronounced differences between samples but no conclusive evidence for differences between taxa [9]. Overall, both dry and wet deposition and contamination of the soil have been shown to contribute to radionuclides in Bryophytes [10]

This pilot study aims to obtain an overview of the uptake of three selected isotopes, ^137^Cs, ^90^Sr and ^241^Am, in mosses from the Chornobyl exclusion zone. All three isotopes are regarded as radionuclides of primary radiological concern [1]. Thereby, it was aimed to include mosses of all major taxonomic groups and from various substrates.

## 2. Materials and Methods

### 2.1. Sampling and Identification of Bryophytes

Bryophyte species were selected in order to cover the greatest possible diversity with regard to taxonomic groups, substrates and growth forms. Samples were collected in paper bags, rinsed with tap water in order to remove soil and dust particles and air-dried. Samples were determined to the species level by morphological and anatomical features [11,12,13]. Macrophotography of mosses was performed with Nikon SMZ 1500, in combination with a Nikon Ds-Ri2 camera. Mosses were partly hydrated, and black velvet was used as a background.

A total of 25 moss samples were collected inside the Chornobyl exclusion zone in October 2018. Sampling focused on the surroundings of the former cooling pond, covering both sediment that has become dry only in recent years [14] and the surrounding woods, at a distance of 3.1–7.5 km from the reactor. Further samples were taken in the abandoned city of Prypiat (Пpип’ять, approximately 3.1–4.2 km from the reactor), and at an open meadow approximately 4.9 km from the reactor, which was also used for radionuclide uptake experiments in the fungus *Schizophyllum commune* [15]. Figure 1 indicates all the sampling locations. Pictures at the sampling locations were taken with an Olympus E3 camera.

### 2.2. γ-Ray Spectroscopy of ^137^Cs and ^241^Am

Both ^137^Cs (T_1/2_ = 30.08 a) and ^241^Am (T_1/2_ = 432.6 a) are γ-ray emitting radionuclides that can be identified and quantified by gamma spectrometry. In this study, a 151 cm^3^ high-purity germanium (HPGe) with a relative efficiency of 35% and a resolution of 1.78 keV (FWHM) at the 1332 keV peak of ^60^Co was used. The software used was Genie 2000. Geometries were calibrated by using QCY48^TM^ solutions for quality assurance. Decay corrections were performed to the date of sampling. Counting time depended on the actual activity of the measured sample but averaged at 99 × 10^3^ s. Uncertainty was calculated via propagation of uncertainty, with one factor taking inhomogeneous distribution of sample material into account. Detection limits were calculated for each sample and nuclide according to DIN 11929.

### 2.3. Quantification of ^90^Sr

Pure β-emitter ^90^Sr (T_1/2_ = 28.91 a) and its decay product ^90^Y (T_1/2_ = 64.05 h) cannot be determined using gamma spectrometry. Their β^-^ radiation was analyzed using liquid scintillation counting (LSC) after chemical isolation of the strontium content of the sample. The entire procedure has been well-described previously [17] and is briefly summarized here: For the microwave-assisted digestion, about 0.2 g of sample material was spiked with 1.2 mg of stable Sr and a known activity of ^85^Sr (for quantification of the extraction yield) and mixed with 5 mL of 8 M HNO_3_. The PTFE vial was closed with a lid and slowly (20 min) heated to 160 °C in a microwave oven. This temperature was held for 30 min. After cooling, the organic components of the sample were entirely dissolved. The sample was filtrated through a Schleicher & Schuell paper filter. The solution was taken up in 3 mL of 8 M HNO_3_ and loaded onto a cartridge that held Eichrom strontium-specific SR-resin. After loading, the resin was rinsed using a solution of 3 M HNO_3_ + 0.05 M oxalic acid. For elution, 10 mL of 0.025 M HNO_3_ was used [18]. For a better miscibility with the LSC cocktail (Ultima Gold AB^TM^), the sample was evaporated to almost dryness and then diluted with 1 mL H_2_O. This procedure was repeated 9 times, which yielded an almost pH-neutral product. After the final evaporation, the sample was transferred to an LSC vial, the flask was rinsed, and the rinse solution also transferred (total volume of sample and rinse solution about 2 mL) and mixed with 18 mL of LSC cocktail. Measurement was performed for 4 × 5 h with the HIDEX 300 SL counter with software Mikrowin 5.58 as described previously [19].

After 14 days, ingrowth of ^90^Y was assumed to have reached secular equilibrium with its mother nuclide. For quality control, the LSC vials were measured again to check if the initial count rate (^90^Sr) had doubled (^90^Sr + ^90^Y).

For quantification, a ^90^Sr standard solution with known activity was used. Decay corrections were performed to the date of sampling.

The detection limit according to DIN 32,645 in each sample was 0.0071 Bq, and the limit to quantify ^90^Sr was 0.0365 Bq.

### 2.4. Statistical Analysis

Stata^®^ 14.2 was used for all analyses. The isotope content measurements were subdivided by taxonomic groups (acrocarpous mosses, pleurocarpous mosses and liverworts) and substrate (asphalt, construction waste, concrete, hortisol with pebbles, podzol, *Dreissena* sp. shells, swamp soil, robinia bark, sand, sandy podzol, sandy mud and gravel). Descriptive statistics comprised arithmetic mean (*μ*), standard deviation (*SD*), quartiles (Q1–Q3) and sample size (*n*). For comparison of subgroups, Kruskal–Wallis Test (with adjustment for ties, KW) with the post hoc Dunn Test (DT) were applied.

The relationship between isotope content and numeric parameters (anatomical features—vascular tissue present, reduced or lacking; distance to Unit 4, fallout from 1986) (200–2000 kBq ^90^Sr/m^2^, 2000–20,000 kBq ^90^Sr/m^2^ and >20,000 kBq ^90^Sr/m^2^) was assessed by linear regressions (LR). Furthermore, regression analyses were also used to confirm the results of the Kruskal–Wallis Test (Appendix A), and to assess the influence of co-founders. In case of activity below the detection limit, 0 was imputed; *p* < 0.05 was regarded as significant.

## 3. Results and Discussion

### 3.1. Moss Diversity in the Study Area

A total of 25 moss samples (17 species) were collected. Two species were Polytrichaceae with functioning vascular systems, ten were more advanced acrocarpous Bryopspidae with sporangia at the apex of the stem and reduced vascular systems, and four were pleurocarpous mosses with lateral sporangia and usually without vascular systems. One species was a thallose liverwort, also without a vascular system. Five species belonged to the genus *Bryum* (Bryaceae). Table 1 shows the main characteristics of the samples and their respective sampling locations. Figure 2 illustrates the diversity of mosses that were collected. Further morphological images are to be found in Appendix A.

Five samples were taken from the sandbanks of the cooling pond, which were formed since 2014 when the pumps maintaining the water level were stopped [14]. Due to constant resuspension of Cs-rich sediments by invertebrates and other mixing processes, especially in shallow water, a comparatively high bioavailability of Cs can be assumed [20]. Three of the moss samples from the pond (*Bryum imbricatum*, *B. argenteum* and *B. badium*, Bryaceae) were collected from low-lying wetlands (54% of the drained bottom area in 2017). The surrounding vegetation consisted of *Populus tremula* (Salicaceae), *Salix* sp. (Salicaceae), *Typha* sp. (Typhaceae), and *Epilobium* sp. (Brassicaceae) (Figure 3f,f’). Two samples (*B.* × *intermedium* and *Marchantia polymorpha*, Marchantiaceae) originated from sandy sites covered with shells (26% of the drained area) (Figure 3c). The mosses were growing directly on *Dreissena* sp. shells under a very loose vegetation of young *Populus*, *Salix* and *Epilobium*.

Eight samples were collected in the woodland west of the cooling pond, dominated by *Robinia pseudacacia* (Fabaceae), *Populus tremula* (Salicaceae), *Quercus* sp. (Fagaceae) and occasional stands of *Pinus sylvestris* (Pinaceae). Substrates included tree bark (*Orthotrichum speciosum,* Orthotrichaceae), construction waste (*Ceratodon purpureus*, Dicranaceae; *Homalothecium philippeanum*, Brachytheciaceae; and *Plagiomnium cuspidatum*, Mniaceae), banquettes (*Brachythecium glaveosum*, Brachiteciaceae), forest floor under *P. sylvestris* (*Dicranum polysetum*, Dicranaceae and *Pleurozium schreberi*, Hypnaceae) and boggy areas (*Amblystegium serpens*, Amblystegiaceae). On the forest floor under deciduous trees, growth of mosses was inhibited by littering.

Eight samples were taken from open habitats, including an oligotrophic meadow (*C. purpureus*, *Polytrichum juniperinum* and *P. piliferum*, Polytrichaceae) (Figure 3b), a sand road on a dike within the cooling pond (*P. piliferum* and *B. caespiticium*) and a former parking space now overgrown by mosses.

Four samples were found in the abandoned city of Prypiat, one in a former garden (*P. juniperinum*) and two growing on concrete (*T. calcicolens* and *D. polysetum*) (Figure 3d), where extensive layers of mosses covered a significant part of the total area. The last sample, *A. serpens*, was growing within an abandoned factory (Figure 3e), where rainwater penetrated the roof, enabling extensive growth of mosses. Furthermore, a cushion of *Climacium dendroides*, Climaciaceae, was found at a road banquette in Prypiat, which could not be sampled due to its extreme activity (dose rate of more than 3 mSv/h at a distance of a few cm), suggesting adsorption of one or more so-called “hot particles” (highly radioactive fuel particles, rich in uranium and fission products [21]).

All collected moss species were typical for their respective habitats [11]. The habitats of 23 samples were exposed to 200–2000 kBq ^90^Sr/m^2^, and two spots in the southern part of Prypiat were contaminated with 2000–20,000 kBq ^90^Sr/m^2^ [1]. No impoverishment of moss diversity due to radiation or other factors was recognizable. Rather, mosses were highly abundant on concrete surfaces, both within Prypiat and throughout the exclusion zone, since weathering does not allow for the growth of vascular plants yet.

### 3.2. ^137^Cesium

^137^Cs was detectable in all moss samples but activities differed widely. Table 2 includes all measured radioisotopes values from 25 sampling sites. An amount of 157–297 Bq/g was found of *M. polymorpha* and *B. intermedium* on *Dreissena* shells, and *B. imbricatum* and *B. badium* on the sand banks of the cooling pond (Figure 3a). *B. argenteum* and *B. caespiticium* exhibited activities > 5 Bq/g, and <1 Bq/g was found exclusively in moss samples growing without direct contact to the substrate, i.e., in epiphytic *O. speciosum*, in mosses growing on construction rubble, and, remarkably, in *D. polysetum* growing on a bollard in the highly contaminated city of Prypiat. These surfaces may not have been exposed to the environment at the time point of the disaster, or ^137^Cs has been washed out over time. *A. serpens* <1 Bq/g was also found growing on the factory floor which was protected against fallout until the roof caved in and rainwater was able to penetrate. Some specimens of the same species (*A. serpens*, *D. polysetum* and *C. purpureus*) were also collected on soil, where they exhibited > 1 Bq/g. Members of the genus *Bryum* generally exhibited a higher activity (KW: *p* = 0.001) compared to the other mosses, in good accordance with the tendency of *Bryum* to accumulate heavy metals as well [22]. No significant difference was found between acrocarpous and pleurocarpous mosses and liverworts, or between mosses with and without vascular tissue. ^137^Cs content differed significantly between the substrates (KW: *p* = 0.029). Shells and sand at the pond were correlated with high ^137^Cs, whereas other substrates were without significant correlations. Deposition in 1986 (LR: ≤2000 vs. >2000 kBq ^90^Sr/m^2^) did not result in a significant difference (KW: *p* = 0.423). Distance to the reactor at Unit 4 was positively correlated with ^137^Cs (LR: F < 0.001; *p* < 0.001; R^2^ = 0.45), however, this is probably an artifact caused by the high ^137^Cs activity at the cooling pond, which was further away from Unit 4 than the other sampling sites.

Except for the high ^137^Cs uptake in *Bryum*, activity seemed to be determined predominantly by the substrate. A previous study in Serbia [23] found substrate to moss ratios of 0.27 to 0.92, but did not identify parameters controlling Cs availability. In the Chornobyl area, pond sediments in shallow waters exhibited the highest ^137^Cs concentrations at the very surface [14], and the very same situation can be assumed for sediments, which have become dry only recently. Furthermore, Cs is strongly adsorbed only to clay minerals, which are virtually absent on the sandy banks of the cooling pond. Thus, it appears plausible that mosses may absorb ^137^Cs from the sediments via their rhizoids, in spite of their very limited capacity for uptake from soil solutions. In terrestrial habitats, ^137^Cs may be less bioavailable, though the sandy podzol soil in the exclusion zone binds Cs only poorly [1]. Furthermore, Cs slowly penetrates into deeper layers of soil and may already be depleted in the very top layer, where it would be accessible for moss rhizoids [21].

### 3.3. ^241^Americium

^241^Am was detectable in 11 of 25 samples, including the specimens collected in the dry meadow (*P. juniperinum*, *P. piliferum* and *C. purpureus*), dike of the cooling pond (*P. piliferum* and *B. caespiticium*), an abandoned parking space (*B. argenteum* and *C. purpureus*) and in the city of Prypiat (*A. serpens*, *T. calcicolens* and *D. polysetum*). ^241^Am was missing from wet habitats and in mosses growing on tree bark or construction waste. Three species exhibited presence or lack of ^241^Am depending on their sampling site (*A. serpens*, *C. purpureus* and *D. polysetum*). No significant influence of fallout in 1986 or distance to Unit 4 was detected, nor was ^241^Am content significantly affected by taxonomic features. KW showed a borderline insignificant (*p* = 0.066) influence of vascular tissues on ^241^Am content. LR, however, indicated increased ^241^Am uptake in species with more developed vascular tissue (F = 0.033; *p* = 0.030; R^2^ = 0.18).

Modeling of ^241^Am accumulation was impeded, since its predecessor ^241^Pu was released in 1986 and continues to decay to ^241^Am, which will achieve its maximum concentration in 2059 [1]. Thus, detection of ^241^Am may indicate availability and uptake of either ^241^Pu or ^241^Am. In contrast to ^137^Cs, ^241^Am uptake is obviously rather independent of the moss species but may involve vascular tissue. In contrast to Cs uptake by potassium carriers, uptake of both Pu and Am by nutrient carriers is very unlikely, since no biological function of actinoids is known. However, metal ions are easily absorbed by a wide variety of biological matrices, e.g., negatively charged cell walls or reduced sulfur [24,25]. Since the whole organism of the moss plant is accessible to Am or Pu due to the lack of an impermeable cuticle, it can be assumed that ^241^Am concentrations reflect the bioavailability of ^241^Am and ^241^Pu in the environment.

In the frame of the disaster, Pu was predominantly dispersed as hot particles and gradually released by weathering; subsequently, soluble metal species may have been relocated by water. In the sediments of the cooling pond, no weathering of hot particles took place, due to reducing, anoxic conditions [26], explaining the lack of ^241^Am content from the moss samples collected at the banks of the cooling pond. ^241^Am is also lacking on substrates that did not exist in 1986, such as tree bark. In soil or on concrete, however, bioavailable ^241^Am is obviously still present in the very top layer, where it is accessible to the rhizoids of mosses. In the case of *A. serpens*, growing within a former factory, hot particles were obviously deposited on the roof. Whereas ^137^Cs was quickly washed out by rain, ^241^Pu/^241^Am seemed to be released continuously until the roof caved in and water released the isotopes unto the factory floor.

### 3.4. ^90^Strontium

^90^Sr was not determined in seven samples. Of these, four exhibited detectable contents (*D. polysetum*, *C. purpureus*, *B. × intermedium* and *B. argenteum*), and three did not (*M. polymorpha*, *B. imbricatum* and *B.* cf. *badium*). As only limited statistical testing was possible in such a small sample, KW did not indicate significant differences (*p* = 0.206). LR, on the other hand, explained ^90^Sr concentration almost completely (LR: R^2^ = 0.99, *p* < 0.001), with significantly higher ^90^Sr on podzol compared to sand or shells. No influence of ^90^Sr fallout in 1986, vascular tissue or taxonomic features was found.

The uptake behavior of ^90^Sr is counterintuitive. Most of the cooling pond received 200–2000 kBq ^90^Sr/m^2^ in 1986, of which approximately 46% was still present after 32 years. As analog for calcium, Sr would be expected to be incorporated into *Dreissena* shells [3], even to such an extent that malformations of the shells have been observed [27]. In 1998, mollusk shells from the cooling pond contained 2500 Bq/kg [28]. However, ^90^Sr uptake may be impeded by the low trophic level of *Dreissena* in the food chain [29]. The ^90^Sr content of the *Dreissena* shells is unknown, but the very low ^90^Sr concentrations in mosses growing directly on the shells suggest that mosses lack acidic exudates as in higher plants, thus no mobilization of ions from the shells occurs, though exudates in general are produced [30]. In podzol, on the other hand, ^90^Sr may also be reversibly bound to Fe and Mn oxides [31,32] and therefore be accessible to moss rhizoids.

## 4. Conclusions

In the Chornobyl exclusion zone, a diverse flora of mosses can be found, with the highest abundance on abandoned streets and on newly formed banks of the drying cooling pond.^137^Cs is generally accumulated by mosses, especially by the genus *Bryum* (Bryaceaea), which is known to accumulate heavy metals. Furthermore, ^137^Cs has been washed out from the very top layer of the soil, where it would be accessible for moss rhizoids. In the newly formed banks of the cooling pond, ^137^Cs is still available for mosses.^241^Am uptake seems to be controlled predominantly by bioavailability, which is higher in podzol than in the cooling pond, where little degradation of hot particles takes place.^90^Sr is taken up by mosses from soil but virtually not from mussel shells, in spite of its similarity to Ca.The presence or lack of functional or reduced vascular tissue in mosses does not seem to influence the uptake of any of the investigated radionuclides.

## Figures and Tables

**Figure 1 toxics-11-00218-f001:**
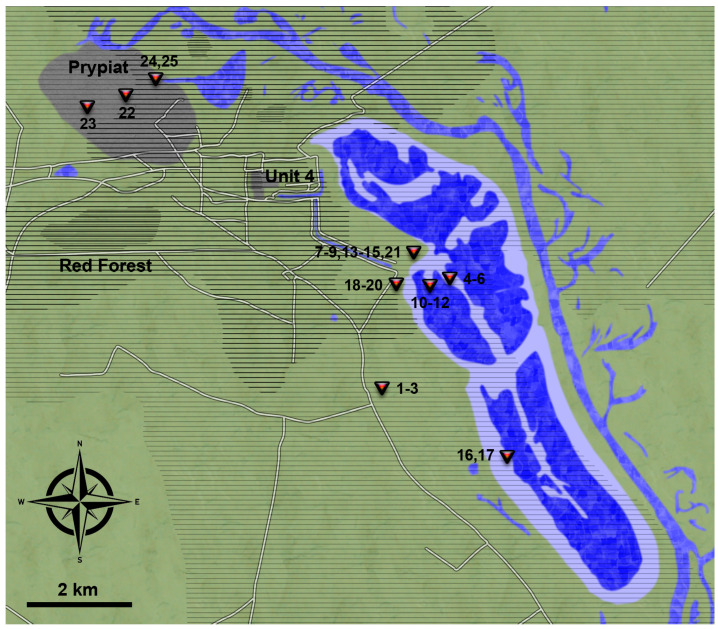
Map of the sampling area. Red pins indicate the sample collection points. In the cooling pond bright blue coloration indicates areas that have recently become dry and dark blue areas were still submerged in 2018. Hatches indicate deposition of 200–2000 kBq ^90^Sr/m^2^, more pronounced, darker hatches 2000–20,000 kBq ^90^Sr/m^2^ and multiple hatches >20,000 kBq ^90^Sr/m^2^. Redrawn after [1,16].

**Figure 2 toxics-11-00218-f002:**
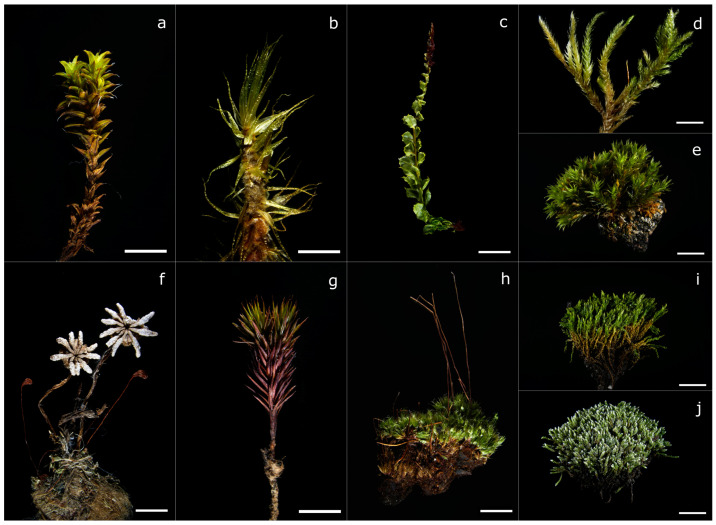
Figure of some of the sampled mosses; # indicates the sampling location, described in detail in Figure 1. (**a**) *Tortula calcicolens* (#24), (**b**) *Dicranum polysetum* (#19), (**c**) *Plagiomnium cuspidatum* (#9), (**d**) *Homalothecium philippeanum* (#8), (**e**) *Orthotrichum speciosum* (#21), (**f**) *Marchantia polymorpha* (#16), (**g**) *Polytrichum piliferum* (#4), (**h**) *Bryum* cf. *badium* (#6), (**i**) *Bryum × intermedium* (#17), (**j**) *Bryum argenteum* (#13). Scale bar = 5 mm.

**Figure 3 toxics-11-00218-f003:**
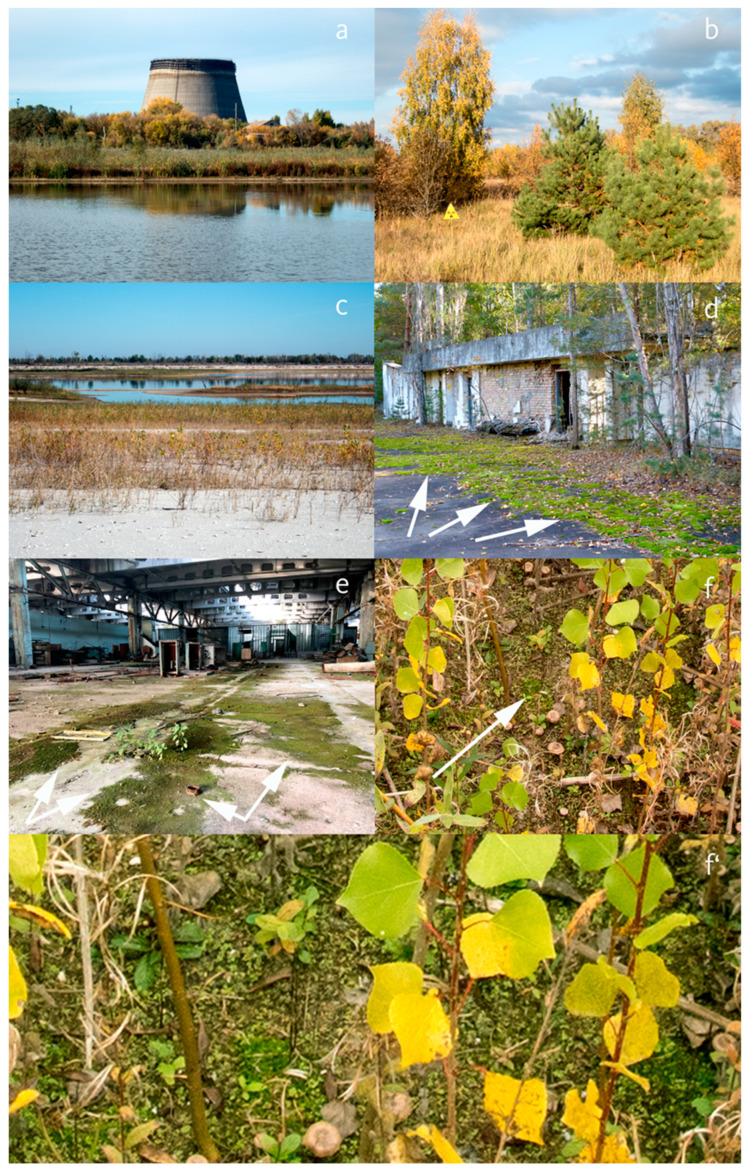
Selected sample sites. (**a**) Cooling pond with unfinished cooling tower, (**b**) dry meadow (#1–3), (**c**) bank of the cooling pond (#16,17), (**d**) Prypiat, moss cushions (#24), (**e**) decaying factory building in Prypiat with *Amblystegium serpens* covering concrete floor (#23), (**f**) *Bryum* moss cushion (#10–12), (**f’**) detail of *Bryum*. Mosses indicated by white arrows.

**Table 1 toxics-11-00218-t001:** Principal characteristics of the Bryophytes used in this study: # indicates the sampling location as described in Figure 1.

ID	Species	Family ^1^	Vascicel ^2^	Position ^3^		Habitat and Substrate
#01	*Polytrichum juniperinum*	Polytrichaceae (acrocarpous)	+	51°20′54.5″ N	30°07′40.9″ E	Open, sandy patches in a meadow, on podzol
#02	*Polytrichum piliferum*	Polytrichaceae (acrocarpous)	+	51°20′54.5″ N	30°07′40.9″ E	Open, sandy patches in a meadow, on podzol
#03	*Ceratodon purpureus*	Dicranaceae (acrocarpous)	±	51°20′54.5″ N	30°07′40.9″ E	Open, sandy patches in a meadow, on podzol
#04	*Polytrichum piliferum*	Polytrichaceae (acrocarpous)	+	51°22′24.9″ N	30°08′36.3″ E	Sandy road on the main dam of the cooling pond
#05	*Polytrichum piliferum*	Polytrichaceae (acrocarpous)	+	51°22′24.3″ N	30°08′36.3″ E	Sandy road on the main dam of the cooling pond
#06	*Bryum caespiticium*	Bryaceae (acrocarpous)	±	51°22′23.4″ N	30°08′37.7″ E	Sandy road on the main dam of the cooling pond
#07	*Ceratodon purpureus*	Dicranaceae (acrocarpous)	±	51°22′28.0″ N	30°08′26.5″ E	Deciduous forest ^4^, on waste material (wood and plastic)
#08	*Homalothecium philippeanum*	Brachytheciaceae (pleurocarpous)	−	51°22′28.0″ N	30°08′26.5″ E	Deciduous forest ^4^, on waste material (wood and plastic)
#09	*Plagiomnium cuspidatum*	Mniaceae (acrocarpous)	±	51°22′28.0″ N	30°08′26.5″ E	Deciduous forest ^4^, on waste material (wood and plastic)
#10	*Bryum imbricatum*	Bryaceae (acrocarpous)	±	51°22′11.5″ N	30°08′26.5″ E	Shrubbery ^5^ on a newly formed island of the cooling pond, on sandy mud
#11	*Bryum argenteum*	Bryaceae (acrocarpous)	±	51°22′11.5″ N	30°08′37.5″ E	Shrubbery ^5^ on a newly formed island of the cooling pond, on sandy mud
#12	*Bryum* cf. *badium*	Bryaceae (acrocarpous)	±	51°22′11.5″ N	30°08′37.5″ E	Shrubbery ^5^ on a newly formed island of the cooling pond, on sandy mud
#13	*Bryum argenteum*	Bryaceae (acrocarpous)	±	51°22′25.1″ N	30°08′19.3″ E	Asphalt parking space, no vegetation besides mosses
#14	*Ceratodon purpureus*	Dicranaceae (acrocarpous)	±	51°22′25.1″ N	30°08′19.3″ E	Asphalt parking space, no vegetation besides mosses
#15	*Brachythecium glaveosum*	Brachytheciaceae (pleurocarpous)	−	51°22′26.0″ N	30°08′19.3″ E	Road side in a deciduous forest ^4^, on gravel
#16	*Marchantia polymorpha*	Marchantiaceae (thallose liverworts)	−	51°20′12.1″ N	30°09′08.1″ E	Shrubbery ^5^ on the newly formed shore of the cooling pond, on shells (*Unio* sp., Unionidae)
#17	*Bryum × intermedium*	Bryaceae (acrocarpous)	±	51°20′12.1″ N	30°09′57.8″ E	Shrubbery ^5^ on the newly formed shore of the cooling pond, on shells (*Unio* sp., Unionidae)
#18	*Pleurozium schreberi*	Hypnaceae (pleurocarpous)	−	51°22′06.9″ N	30°08′05.7″ E	Open forest dominated by *Pinus sylvestris* (Pinaceae), on raw humus
#19	*Dicranum polysetum*	Dicranaceae (acrocarpous)	±	51°22′06.9″ N	30°08′05.7″ E	Open forest dominated by *Pinus sylvestris* (Pinaceae), on raw humus
#20	*Amblystegium serpens*	Amblystegiaceae (pleurocarpous)	−	51°22′07.2″ N	30°08′05.7″ E	Swampy meadow, between grasses
#21	*Orthotrichum speciosum*	Orthotrichaceae (acrocarpous)	−	51°22′27.0″ N	30°08′05.3″ E	Deciduous forest ^4^, on the bark of *Robinia pseudacacia* (Fabaceae), 2 m above the ground
#22	*Polytrichum juniperinum*	Polytrichaceae (acrocarpous)	+	51°24′16.9″ N	30°03′13.2″ E	Former garden of a housing complex in Prypiat, on gravel and garden soil
#23	*Amblystegium serpens*	Amblystegiaceae (pleurocarpous)	−	51°24′08.0″ N	30°03′13.2″ E	Concrete floor of a decaying factory building in Prypiat
#24	*Tortula calcicolens*	Pottiaceae (acrocarpous)	±	51°24′27.4″ N	30°02′31.6″ E	Asphalt parking space in Prypiat, no vegetation besides mosses
#25	*Dicranum polysetum*	Dicranaceae (acrocarpous)	±	51°24′26.7″ N	30°03′53.8″ E	Concrete bollard in Prypiat

^1^ Families are itemized to thallose liverworts (no leafy liverworts were found), primitive acrocarpous mosses with sporangia at the apex of the stem and an existing but not necessarily functioning vascular system, and more advanced pleurocarpous mosses with lateral sporangia and usually without a vascular system. ^2^ Vascular system: + present and functional; ± reduced; − missing. ^3^ Altitude was virtually constant, between 96 and 120 m.a.s.l. ^4^ The deciduous forests at the cooling pond are dominated by *Robinia pseudacacia* (Fabaceae) and *Populus tremula* (Salicaceae) with some intermixed *Quercus petraea* (Fagaceae). ^5^ The shrubbery at the cooling pond is dominated by *Populus tremula* (Salicaceae), *Salix* sp. (Salicaceae), *Typha* sp. (Typhaceae) and *Epilobium* sp. (Brassicaceae).

**Table 2 toxics-11-00218-t002:** Concentration of selected radionuclide in bryophytes [Bq/g]; # indicates the sampling location as described in Figure 1.

ID	Species	^137^Cs		^241^Am		^90^Sr (Uncertainty)
		Radionuclides (Uncertainty)	Detection Limit	Radionuclides (Uncertainty)	Detection Limit	
#01	*Polytrichum juniperinum*	6.0 (5.9–8.4)	0.402	0.24 (0.23–0.34)	0.035	Not determined
#02	*Polytrichum piliferum*	3.6 (2.9–4.2)	0.208	0.14 (0.12–0.17)	0.04	Not determined
#03	*Ceratodon purpureus*	5.5 (4.9–6.9)	0.309	0.22 (0.19–0.29)	0.044	Not determined
#04	*Polytrichum piliferum*	3.1 (2.7–3.4)	0.450	0.08 < A < 0.17 ^1^	0.171	Not determined
#05	*Polytrichum piliferum*	6.5 (5.6–6.9)	0.758	0.31 (0.25–0.35)	0.127	Not determined
#06	*Bryum caespiticium*	10.9 (9.6–11.5)	0.325	0.43 (0.37–0.46)	0.066	Not determined
#07	*Ceratodon purpureus*	0.6 (0.5–0.7)	0.053	<0.001 ^2^	4.36 × 10^−4^	Not determined
#08	*Homalothecium philippeanum*	0.9 (0.9–1.2)	0.126	<0.001 ^2^	8.86 × 10^−4^	Not determined
#09	*Plagiomnium cuspidatum*	0.6 (0.6–0.8)	0.107	<0.001 ^2^	5.12 × 10^−4^	Not determined
#10	*Bryum imbricatum*	248 (219–261)	6.49	<0.011 ^2^	0.021	<0.45 ^2^
#11	*Bryum argenteum*	37.1 (30.8–43.4)	1.20	<0.003 ^2^	4.32 × 10^−3^	0.64 (0.63–0.65)
#12	*Bryum* cf. *badium*	157 (136.4–166.1)	3.63	<0.006 ^2^	0.011	<0.38 ^2^
#13	*Bryum argenteum*	8.4 (6.9–9.9)	0.278	0.28 (0.23–0.34)	0.067	2.54 (2.53–2.55)
#14	*Ceratodon purpureus*	3.2 (2.8–3.4)	0.138	0.06 (0–0.08)	0.039	Not determined
#15	*Brachythecium glaveosum*	2.3 (2.1–2.9)	0.247	<0.022 ^2^	0.077	Not determined
#16	*Marchantia polymorpha*	185 (154–216)	5.90	<0.011 ^2^	0.021	<0.47 ^2^
#17	*Bryum × intermedium*	297 (259–317)	9.04	<0.017 ^2^	0.034	0.76 (0.75–0.77)
#18	*Pleurozium schreberi*	3.0 (2.7–3.9)	0.289	<0.001 ^2^	7.53 × 10^−4^	Not determined
#19	*Dicranum polysetum*	15.9 (13.3–18.5)	1.22	<0.004 ^2^	8.00 × 10^−3^	12.14 (12.13–12.15)
#20	*Amblystegium serpens*	2.9 (2.4–3.4)	0.244	<0.002 ^2^	2.11 × 10^−3^	Not determined
#21	*Orthotrichum speciosum*	0.8 (0.8–1)	0.069	<0.001 ^2^	3.60 × 10^−4^	Not determined
#22	*Polytrichum juniperinum*	3.0 (2.4–3.6)	0.276	<0.001 ^2^	1.57 × 10^−3^	Not determined
#23	*Amblystegium serpens*	0.9 (0.8–1.2)	0.109	0.04 (0.03–0.05)	0.026	Not determined
#24	*Tortula calcicolens*	4.8 (4.1–5)	0.627	0.32 (0.27–0.36)	0.107	Not determined
#25	*Dicranum polysetum*	0.8 (0.8–1.1)	0.068	0.015 (0.01–0.02)	0.013	Not determined

^1^ Detectable but not quantifiable. ^2^ Below detection limit.

## Data Availability

Data will be made available upon request.

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
