# Peer review of "Uptake of Radionuclides by Bryophytes in the Chornobyl Exclusion Zone"

_toxics, 2023, doi:10.3390/toxics11030218_

Round 1

Reviewer 1 Report

The paper "Uptake of radionuclides by Bryophytes in the Chernobyl Exclusion zone"  is well written work with clear aim, adequate methods and appropriate conclusions. It is written in good English language. 

The minor revision is required. Namely the chapter (2.2.) on gamma-ray spectrometry needs to be extended with more detail. The authors need to add: average or exact counting time, detection limits for each of the radionuclides of interest, quality assurance data and software used. 

Author Response

Dear Reviewer 1,

Thank you very much for your valuable comments and beneficial suggestions. The original manuscript was changed accordingly and now sounds a lot clearer and more precise. Suggested changes and answers to comments are in blue colour, additionally, the alterations to the original manuscript were added in smaller print (11 instead of 12 like the rest of the document).

Thank you for your thoughts and helpful suggestions!

Kind regards from the author team

Reviewer 2 Report

#1 General comment: This is a detailed study of the uptake of radionuclides including Am-241by Bryophytes in the vicinity of a site where a severe nuclear accident occurred. As the authors pointed out, since Am-241 is produced from Pu-241, the amount of Am-241 in the environment originating from this accident will reach a maximum in 2059, and its radioactivity will be 43 times compared to the time of the accident, so it is necessary to observe the transition.

#2 line 153: Results of the logistic regression analysis are not shown.

#3 line 216: “does rate” seems to be “dose rate”

#4 line 217: Would the accumulation of hot particles cause the dose rate to be so high?

#5 line 242: Since the results of the regression analysis are not shown, it would be better to indicate what model was used.

#6 line 273: Modeling of Am-241 accumulation may be possible with time series data, since it will be a combination of supply from the preceding Pu-241 and the remaining released Am-241 in conjunction with the respective release behavior of Pu and Am into the environment as described subsequently.

#7 Table 2:Uncertanitiy” would be a typo for “Uncertanity”. A range is shown as uncertainty, but it would be better to indicate the definition of the uncertainty.

#8 line 367: "Agency, I.A.E." would be better described as "International Atomic Energy Agency".

Author Response

Dear Reviewer 2,

Thank you very much for your valuable comments and beneficial suggestions. The original manuscript was changed accordingly and now sounds a lot clearer and more precise. Suggested changes and answers to comments are in blue colour, additionally, the alterations to the original manuscript were added in smaller print (11 instead of 12 like the rest of the document).

Thank you for your thoughts and helpful suggestions!

Kind regards from the author team

Reviewer 3 Report

Paper describes activity concentration of Cs, Sr, Am in Chornobyl area. There are many papers about this topic in literature, so what is novelty of proposed study? Discussion have to be enhanced by comparison with previously studies. More information about accumulation of radionuclides in mosses should be added in introduction section. In Table 1 column in position should be removed. Statistical analysis is hard to understand. Kruskal-Wallis analysis of variance with Dunn's test is very strong chemometric tool, so for what reason Authors additionaly applied e.g. Mann-Whitney test? If there are some significant independences, applied K-W ANOVA and Dunn's test reveal it. In conclusion there is information about taken up from shells, was it goal of presented study? 

Author Response

Dear Reviewer 3,

Thank you very much for your valuable comments and beneficial suggestions. The original manuscript was changed accordingly and now sounds a lot clearer and more precise. Suggested changes and answers to comments are in blue colour, additionally, the alterations to the original manuscript were added in smaller print (11 instead of 12 like the rest of the document).

Thank you for your thoughts and helpful suggestions!

Kind regards from the author team

Round 2

Reviewer 3 Report

Dear Authors, I can not agree with your explanation about my comments. You should know that Kruskal-Wallis test is commonly named "nonparametric anova", so you used anova. More over with KW test you used Dunn's test- if there are differences between groups this post-hoc test reveal which groups different significantly. I can not aree that Mann-Whitney test should be additionaly use to compare pairs/groups. It better to use one strong statistical test, than use few test to find correlations (even, if there are no correlations). 

Author Response

Dear Reviewer,

We carefully implemented your suggestions.

Kind regards from the author team

Open Review

English language and style

( ) English very difficult to understand/incomprehensible
( ) Extensive editing of English language and style required
( ) Moderate English changes required
(x) English language and style are fine/minor spell check required
( ) I don't feel qualified to judge about the English language and style

Yes

Can be improved

Must be improved

Not applicable

Does the introduction provide sufficient background and include all relevant references?

( )

( )

(x)

( )

Are all the cited references relevant to the research?

( )

( )

(x)

( )

Is the research design appropriate?

( )

( )

(x)

( )

Are the methods adequately described?

( )

( )

(x)

( )

Are the results clearly presented?

( )

( )

(x)

( )

Are the conclusions supported by the results?

( )

( )

(x)

( )

Comments and Suggestions for Authors

Dear Authors, I can not agree with your explanation about my comments. You should know that Kruskal-Wallis test is commonly named "nonparametric anova", so you used anova. More over with KW test you used Dunn's test- if there are differences between groups this post-hoc test reveal which groups different significantly. I can not aree that Mann-Whitney test should be additionaly use to compare pairs/groups. It better to use one strong statistical test, than use few test to find correlations (even, if there are no correlations). 

Authors respond: Though we still believe that regression analysis is a valid tool for the analysis of our data, we see your point that presenting the p-values of the Kruskal-Wallis test may facilitate reading of the paper. We therefore implemented the Kruskal-Wallis test as our primary tool and rephrased the material and methods section accordingly. Since the Kruskal-Wallis test with 2 groups is equivalent to the Mann-Whitney test, we now use the Kruskal-Wallis test thoughout the manuscript. However, the Kruskal-Wallis test was poorly suitable to asses the relationship between metric parameters like the distance to the reactor or also the development of vascular tissue. In these cases regression was still used. The output of the regression analysis has been included in a supplement table.

Submission Date

03 January 2023

Date of this review

01 Feb 2023 15:13:24

Round 3

Reviewer 3 Report

The article has been revised accordingly by the reviewer comments. Thank you